# Robust Validation and Comprehensive Analysis of a Novel Signature Derived from Crucial Metabolic Pathways of Pancreatic Ductal Adenocarcinoma

**DOI:** 10.3390/cancers14071825

**Published:** 2022-04-04

**Authors:** Wenchao Gu, Shaocong Mo, Yulin Wang, Reika Kawabata-Iwakawa, Wei Zhang, Zongcheng Yang, Chenyu Sun, Yoshito Tsushima, Huaxiang Xu, Takahito Nakajima

**Affiliations:** 1Department of Diagnostic Radiology and Nuclear Medicine, Gunma University Graduate School of Medicine, Maebashi 371-8511, Japan; guwc@gunma-u.ac.jp (W.G.); yoshitotsushima@gunma-u.ac.jp (Y.T.); 2Department of Diagnostic and Interventional Radiology, University of Tsukuba, Ibaraki 305-8577, Japan; 3Department of Digestive Diseases, Huashan Hospital, Fudan University, Shanghai 200437, China; scmo16@fudan.edu.cn; 4Department of Nephrology, Zhongshan Hospital, Fudan University, Shanghai 200437, China; 16301020044@fudan.edu.cn; 5Division of Integrated Oncology Research, Gunma University Initiative for Advanced Research, Maebashi 371-8511, Japan; r.kawabata@gunma-u.ac.jp; 6Division of Cancer Sciences, School of Medical Sciences, Faculty of Biology, Medicine and Health, University of Manchester, Manchester M13 9PL, UK; wei.zhang-22@postgrad.manchester.ac.uk; 7Division of Life Sciences and Medicine, Center of Stomatology, The First Affiliated Hospital of USTC, University of Science and Technology of China, Hefei 230026, China; yang_chen1107@ustc.edu.cn; 8AMITA Health St. Joseph Hospital, Chicago, IL 60657, USA; chenyu.sun@amitahealth.org; 9Department of Pancreatic Surgery, Shanghai Cancer Center, Fudan University, Shanghai 200437, China

**Keywords:** pancreatic ductal adenocarcinoma, metabolic pathways, prognosis, recurrence, drug sensitivity

## Abstract

**Simple Summary:**

Pancreatic ductal adenocarcinoma (PDAC) has a high mortality rate and a poor prognosis. To solve the above limitations of multiomics studies of metabolism in PDAC and optimize the prognosis of PDAC clinically, we demonstrated a 16-gene prognostic signature based on the metabolic pathways called gbcxMRS. The prognostic value varied in six public datasets and our own data cohort in Shanghai Cancer Center by RT-PCR. Notably, gbcxMRS also accurately predicted poor PDAC subtypes and recurrence. It also highly associated with immune infiltration cells. Furthermore, high gbcxMRS may indicate high sensitivity to irinotecan, docetaxel, and CTLA4 inhibitor immunotherapy.

**Abstract:**

Pancreatic ductal adenocarcinoma (PDAC) is a malignant tumor with a dismal prognosis. PDAC have extensively reprogrammed metabolic characteristics influenced by interactions with normal cells, the effects of the tumor microenvironment and oncogene-mediated cell-autonomous pathways. In this study, we found that among all cancer hallmarks, metabolism played an important role in PDAC. Subsequently, a 16-gene prognostic signature was established with genes derived from crucial metabolic pathways, including glycolysis, bile acid metabolism, cholesterol homeostasis and xenobiotic metabolism (gbcx). The signature was used to distinguish overall survival in multiple cohorts from public datasets as well as a validation cohort followed up by us at Shanghai Cancer Center. Notably, the gbcx-related risk score (gbcxMRS) also accurately predicted poor PDAC subtypes, such as pure-basal-like and squamous types. At the same time, it also predicted PDAC recurrence. The gbcxMRS was also associated with immune cells, especially CD8 T cells, Treg cells. Furthermore, a high gbcxMRS may indicate high drug sensitivity to irinotecan and docetaxel and CTLA4 inhibitor immunotherapy. Taken together, these results indicate a robust and reproducible metabolic-related signature based on analysis of the overall pathogenesis of pancreatic cancer, which may have excellent prognostic and therapeutic implications for PDAC.

## 1. Introduction

Pancreatic cancer is one of the most aggressive gastrointestinal tumors and ranks as the fourth leading cause of cancer mortality in the USA and the sixth leading cause of cancer mortality in China [1,2]. The World Health Organization reported 458,918 new cases of pancreatic cancer in 2018, representing 2.5% of all cancers [3]. Additionally, 432,242 deaths due to pancreatic cancer in the same year were reported, accounting for 4.5% of all cancers [4]. Being male and being older as well as being a smoker and having a family history of pancreatic cancer are widely recognized as risk factors associated with an increased incidence and mortality of pancreatic cancer [5,6]. Due to the lack of early clinical symptoms, more than 80% of patients with pancreatic cancer present with advanced-stage diagnosis, when it is too late to undergo surgery for tumor resection [7]. In addition, the current chemotherapeutic regimen is limited and usually unsatisfactory [8,9,10]. Compared with patients with the same cancer stage who do not undergo surgery, those who do undergo surgery may experience an additional 10 months or more of life, although they will also likely experience indisposition and recurrence [8,11]. Hence, early diagnosis and timely treatment are associated with a reduced incidence and mortality of pancreatic cancer.

Pancreatic cancer mainly consists of two pathological types, pancreatic ductal adenocarcinoma (PDAC) and pancreatic endocrine tumors. PDAC accounts for the majority of pancreatic cancer cases (>85% of all cases), while pancreatic endocrine tumors represent <5% [12,13,14]. Pancreatic cancer cells are tumor cells that have extensively reprogrammed metabolic characteristics influenced by interactions with normal cells, the effects of the tumor microenvironment (TME) and oncogene-mediated cell-autonomous pathways [15,16,17]. Glycolysis can be induced by hypoxia and is reported to promote tumor progression and chemoresistance in PDAC [18,19,20]. Pancreatic cancer cells tend to express more facilitated transporters (GLUTs) and symporters (SGLTs), two types of glucose transporters; therefore, tumor cells can take up more glucose than other noncancer pancreatic tissues [21,22]. In addition, the feedback system of glycolysis in PDAC is disordered because two lactate transporters, monocarboxylate transporter 1 (MCT1) and monocarboxylate transporter 4 (MCT4), are also overexpressed to allow tumor cells to transport the gathered lactate outside the cell [19,23]. The association between bile acids and PDAC has been known for decades [24]. Because of a common duct, the increased bile acids can reflux into the pancreatic duct and induce the transformation of epithelial cells or acinar cells into PDAC cells [25]. In addition, bile acids have been confirmed to stimulate matrix metalloproteinase (MMP) production, a protein family involved mainly in the breakdown of extracellular matrix, which can enhance the aggressive ability of pancreatic cancer cells [26]. Interestingly, the epidermal growth factor receptor (EGFR) family can also be activated, and bile acids induce the overexpression of ERBB2 (HER2) in the tumor tissue contacted, leading to a significantly worse prognosis [27,28,29,30]. In PDAC tumor cells, the loss of the tumor suppressor *TP53*, which regulates metabolism and energy intake in cells, combined with oncogenic *KRAS* mutations, has been shown to enhance the uptake and consumption of cholesterol [31,32]. Nevertheless, whether to inhibit the cholesterol pathway using statins is still controversial [33,34,35,36,37]. The distinct responses to statins may result from the different characteristics of the tumor cells [33,38].

Recent advances have revealed that high-throughput next-generation sequencing technology and gene chips can provide abundant prognostic information for PDAC, based on which many studies have been carried out to construct signatures to predict the overall survival (OS) of PDAC patients [39,40,41]. Yan et al. defined a gene signature consisting of *LYRM1*, *KNTC1*, *IGF2BP2* and *CDC6* that was significantly associated with the progression and prognosis of PDAC [42]. Furthermore, Tan et al. focused on the reprogramming of glycolysis and lipid metabolism in PDAC. They established a three-gene signature including *MET*, *ENO3* and *CD36* to estimate and assess the OS of PDAC patients [43]. However, these predictive signatures for PDAC have some limitations. First, the significance of metabolism in PDAC has not been fully demonstrated. Second, the functional mechanism of these signatures has not been revealed [44].

To solve the above limitations of multiomics studies of metabolism in PDAC and optimize the prognosis of PDAC clinically, we first highlighted the role of metabolism in tumor progression in PDAC and discovered the most crucial metabolic pathways. Next, a robust prognostic signature was established using the genes in these pathways (glycolysis, bile acid metabolism, cholesterol homeostasis and xenobiotic metabolism-related risk score, gbcxMRS), which underwent repetitive validation. We observed that gbcxMRS predicted poor OS and poor subtypes of PDAC as defined by previous studies. Subsequently, we demonstrated the underlying mechanisms, tumor microenvironment and drug sensitives related to the 16-gene signature in detail. Taken together, our signature permits a notably better prognosis and understanding of PDAC.

## 2. Materials and Methods

### 2.1. Patient Cohort

As demonstrated in the flow chart (Figure 1), E-MTAB-6134 from the ArrayExpress database (https://www.ebi.ac.uk/arrayexpress/, accessed on 22 March 2021) was used as the training cohort, which contained 288 patients with PDAC. PACA-AU from the International Cancer Genome Consortium (ICGC, https://dcc.icgc.org/, accessed on 22 March 2021) was used as the primary validation cohort, and only patients whose pathological type was PDAC were included (*n* = 66). In addition, GSE79668 (*n* = 51), GSE71729 (*n* = 123), GSE62452 (*n* = 65) and GSE28735 (*n* = 42) were obtained from the Gene Expression Omnibus (GEO, https://www.ncbi.nlm.nih.gov/geo/, accessed on 16 March 2021) as 4 secondary validation cohorts [45,46,47,48]. For each GEO dataset, only primary PDAC tissues remained. The baseline and phenotypic information of the datasets were collected (Appendix A). For RNA sequencing, the expression profile in the format of transcripts per kilobase of exon model per million mapped reads (TPM) was used. The detail information of each dataset are summarized in Appendix A.

Furthermore, we enrolled 34 patients with PDAC who underwent radical resection from 5 September to 28 December 2015, at Shanghai Cancer Center, Fudan University (FUSCC). Fresh frozen tissues were obtained from surgical specimens. According to the standardized strategy, each patient underwent routine follow-up (every 3 months until the patient died). Overall survival (OS) was defined as the interval from the surgical day to death or the last follow-up. 

### 2.2. Gene Set Enrichment Analysis

To evaluate the metabolic impact on PDAC, 50 hallmark gene sets were obtained from Msigdb via the EnrichmentBrowser package [49]. Gene set variation analysis (GSVA) was subsequently applied to conduct single-sample gene set enrichment analysis (ssGSEA), after which 0–1 normalization and scale normalization were conducted [50]. Other gene signatures were obtained from the “IOBR” package. For gene set enrichment analysis (GSEA), genes were ranked either by the fold change between two groups or by the correlation with the gbcxMRS. ClusterProfiler was used to conduct GSEA with gene sets from Kyoto Encyclopedia of Genes and Genomes (KEGG), with visualizations generated using gseaplot2 [51]. In addition, differentially expressed genes (DEGs) were screened out between the high and low gbcxMRS groups with |log2Fold Change| ≥ 1 and adjusted *p* value < 0.05 via the limma package. ClusterProfiler was also used for the Gene Ontology analysis of the DEGs.

### 2.3. Consensus Clustering

Genes were extracted from crucial metabolic pathways, after which consensus clustering was implemented on the metabolic expression profile via the ConsensusClusterPlus package, to perform the distance calculation by using the “Partition Around Medoids” algorithm [52]. The optimal number of clusters was determined by an empirical cumulative distribution function (CDF).

### 2.4. Immune Cell Infiltration Evaluation

To evaluate the relative expression of immune cells in the TME, GSVA and ssGSEA were conducted with immune cell signatures obtained from a previous study [53]. Then, immune cell infiltration between the high and low gbcxMRS groups was assessed with the Wilcoxon test.

### 2.5. Drug Sensitivity Analysis

The drug sensitivity data for cancer cell lines were obtained from the Cancer Therapeutics Response Portal (CTRP v2.0, https://portals.broadinstitute.org/ctrp, accessed on 27 April 2021), the PRISM Repurposing dataset (PRISM, https://depmap.org/portal/prism/, accessed on 27 April 2021) and CellMiner (http://discover.nci.nih.gov/cellminer, accessed on 27 April 2021) [54,55]. The dose–response curve (area under the curve: log-AUC) value was estimated in the gbcxMRS groups for drug sensitivity using ridge regression [56,57]. Subclass mapping was used to predict the response to immune checkpoint blockade with GenePattern (https://www.genepattern.org/, accessed on 27 April 2021) [58]. Subsequently, the RNA-sequencing profile and the drug sensitivity data of 60 cell lines (NCI-60) were downloaded from CellMiner. Chemotherapy drugs for PDAC, such as gemcitabine, oxaliplatin, irinotecan, etc., were selected. The K-nearest neighbor (KNN) algorithm was utilized to impute missing data. Then, the correlations between the 16 genes and the drug sensitivities were calculated. The half maximal inhibitory concentration (IC50) was used to evaluate the drug sensitivity between gbcxMRS groups. 

Section 2.6 shows the quantification of mRNA expression levels by quantitative real-time PCR.

Total RNA of each of the 34 samples was extracted by TriPure Isolation reagent (Roche, Shanghai, China). Complementary DNA was acquired using the M-MLV Reverse Transcriptase Synthesis kit (Promega, WI, USA). qRT-PCR was conducted with the Power SYBR Green PCR Master Mix kit (Applied Biosystems, CA, USA). Subsequently, relative transcript expression was calculated by the ΔΔCt method. ACTB (β-actin) was applied as the endogenous reference. The primer sequences used in the qRT-PCR are listed in Appendix A.

### 2.6. Statistical Analysis

Univariate Cox regression was used to filter the factors with prognostic values (*p* value < 0.05), including hallmark pathways, metabolic genes and clinical variants. Variants were shrunk by the least absolute shrinkage and selection operator (LASSO) and stepwise regression. Multivariate Cox regression was applied to construct the metabolic signature and combine the gbcxMRS with other clinical variants. Kaplan–Meier analysis was used to distinguish the clinical outcomes (log-rank *p* value < 0.05). The area under the receiver operating characteristic curve (ROC-AUC) was utilized to assess the predictive value of the gbcxMRS. In addition, survival-related predictions were measured by the time-dependent ROC curve. Correlations were measured by the Spearman method, and comparisons between groups were performed by the Wilcoxon method. Comparisons between the upper and lower quartiles were performed when necessary. All statistical analyses were performed in R (version 4.0.3).

## 3. Results

### 3.1. Key Metabolic Pathways of PDAC and Molecular Subtyping

The text continues here. To investigate the key metabolic pathways in PDAC, single sample gene set enrichment analysis was first applied in the E-MTAB-6134 cohort (*n* = 288) with hallmark gene sets. Ten pathways were identified as prognostic factors after univariate Cox regression, among which four pathways were closely related to metabolism, including glycolysis (HR: 2.987), bile acid metabolism (HR: 0.368), cholesterol homeostasis (HR: 1.656) and xenobiotic metabolism (HR: 0.493) (Figure 2a). Next, a total of 545 genes extracted from the 4 pathways were integrated to screen out the metabolic subtypes in PDAC. Consensus clustering assigned the 288 patients into 2 distinct subtypes (Figure 2b). A heatmap of the enrichment score and GSEA confirmed the metabolic differences between the two subtypes (Figure 2c,d). Notably, we observed that Subtype 2 had a worse prognosis (log-rank *p* value = 0.00031) (Figure 2e). To validate the subtyping, the ICGC cohort was utilized to conduct consensus clustering, whereby the patients were divided into six clusters (Appendix A). Cluster 1 (*n* = 17) and Cluster 3 (*n* = 13) had similar patterns to those of two subtypes in E-MTAB-6134, with Cluster 3 exhibiting a relatively worse survival (Figure 2f,g). Together, glycolysis, bile acid metabolism, cholesterol homeostasis and xenobiotic metabolism impacted PDAC.

### 3.2. Development of the Signature from Key Metabolic Pathways in E-MTAB-6134

To identify a metabolism-related prognostic signature for PDAC, a total of 520 genes involved in glycolysis, bile acid metabolism, cholesterol homeostasis and xenobiotic metabolism first underwent univariate Cox regression, which screened 142 genes with prognostic value. LASSO-Cox regression was used to shrink the variants to 34, after which stepwise multivariate regression identified 16 genes with which to construct the signature (Figure 3a). Among the 16 genes, 5 were involved in glycolysis, 3 were involved in bile acid metabolism, 2 were involved in cholesterol homeostasis, and 6 were involved in the xenobiotic metabolism pathway (Figure 3b, Appendix A). Subsequently, the risk score, termed the gbcxMRS of each patient in the E-MTAB-6134 cohort, was calculated. It was reassuring to note that the signature could distinguish the survival status, and high gbcxMRS patients tended to have a worse prognosis (Figure 3c,d). In addition, a time-dependent ROC curve exhibited high efficacy in predicting OS (Figure 3e).

### 3.3. Robust and Repeated Validation of the 16-Gene Signature in External Cohorts

To illustrate the significance of our signature, we sought to repeatedly validate the signature with multiple external datasets, including the FUSCC cohort from our hospital. The PACA-AU cohort from the ICGC database (ICGC cohort) was used as a primary validation cohort, and GSE79668, GSE71729, GSE62452 and GSE28735 were secondary validation cohorts. Notably, the signature could distinguish the OS in the ICGC cohort with relatively high efficacy (Figure 4a–c). The OS of GSE79668, GSE71729 and GSE62452 could also be distinguished by gbcxMRS using Kaplan–Meier analysis (Figure 4d–f). Although there was insufficient evidence to demonstrate the survival difference in GSE28735 (*n* = 42), the signature still worked for the subgroup of patients whose survival time was less than 3 years (Appendix A). More importantly, the gbcxMRS was also capable of distinguishing the OS of the FUSCC cohort (log-rank *p* value = 0.04) (Figure 4g). The AUC value of gbcxMRS for the prediction of recurrence was 78.7% (Figure 4h). The gbcxMRS was able to make similarly accurate predictions (AUC:71.9%) for recurrence in the ICGC cohort (Figure 4i). Overall, we demonstrated that the 16-gene signature could be applied to external cohorts through robust validation.

### 3.4. GbcxMRS Was an Independent and Indispensable Prognostic Factor in PDAC

To verify that gbcxMRS was a prognostic independent factor from other clinical variants, multivariate Cox analysis was conducted on the training and validation cohorts. The gbcxMRS was significantly independent of sex, staging or mutations, etc., in the training cohort (Figure 5a). Similar results were obtained in the cohorts with detailed clinical phenotypes (Figure 5b–d). In addition, the baseline data of the high and low gbcxMRS groups were compared in the training cohort, which showed that the high gbcxMRS patients tended to have higher grades (Table 1). Furthermore, interestingly, combined with other independent prognostic variants, the gbcxMRS even showed a worse performance in time-dependent ROC curve analysis, which indicated that gbcxMRS had indispensable predictive value (Figure 5e). In addition, we also found that gbcxMRS was significantly associated with recurrence, CA199, CA242 and CEA in our FUSCC cohort (Table 2).

### 3.5. Functional Enrichment for the gbcxMRS

To explore the mechanism by which gbcxMRS was associated with high-risk patients, we first screened out the DEGs between the high- and low-gbcxMRS groups in E-MTAB-6134 (upper and lower quantile gbcxMRS patients were included). Twenty-seven upregulated and 58 downregulated DEGs were identified (|log2Fold change| < 1, adjusted *p* value < 0.05; Appendix A). GO analysis revealed that upregulated genes were mainly involved in extracellular matrix (ECM) organization, while downregulated genes were involved in tissue homeostasis (Figure 6a,b; Appendix A). For the ICGC cohort, 130 upregulated DEGs were found, which were also involved in extracellular matrix organization, while 334 downregulated genes mainly participated in various metabolic pathways, as expected (Figure 6c,d; Appendix A). To verify this result, the correlation between each gene and gbcxMRS was calculated, after which the genes were input into GSEA and ranked by the correlation coefficient. In the E-MTAB-6134 cohort, proteasome, spliceosome and ECM receptor interactions were activated as gbcxMRS increased, while peroxisome and cell adhesion molecular (CAM) marker pathways were suppressed (Appendix A and Figure 6e,f; Appendix A). Similar results were obtained in the ICGC cohort (Appendix A; Appendix A). Additionally, considering that gbcxMRS suppressed peroxisome pathway activity, we further discovered that the high gbcxMRS group had a higher ssGSEA score for ferroptosis, an important form of programmed cell death (Appendix A). Taken together, the results indicate that the gbcxMRS interacted with various pathways, among which ECM organization tended to be significant.

### 3.6. GbcxMRS Predicted PDAC Subtypes and Influenced the TME

Some studies have reported subtypes of PDAC with poor prognosis when stratification was conducted, such as the pure basal-like subtype based on the E-MTAB-6134 cohort and the squamous type based on the ICGC cohort [59,60]. To investigate whether gbcxMRS could predict the subtypes of PDAC with poor prognosis, we first screened out the different distribution patterns of subtypes in high and low gbcxMRS patients in both the E-MTAB-6134 and ICGC cohorts (Figure 7a,b). Apparently, subtypes with poor prognosis were more frequently distributed among the high gbcxMRS patients. Strikingly, gbcxMRS robustly predicted the pure basal-like type in E-MTAB-6134 (AUC = 86.6%, Figure 7c, left panel) and the squamous type in the ICGC cohort (AUC = 89.7%, Figure 7c, right panel). Additionally, gbcxMRS predicted the immune classical type, a benign subtype of PDAC, with relatively high accuracy (AUC = 73%, Figure 7c, middle panel). Considering the association between gbcxMRS and immunity, ssGSEA was applied to reveal the impact of gbcxMRS on immune cells in the TME, which showed that the high gbcxMRS group tended to have a lower infiltration of CD8 T cells and B cells and higher Treg cells (Figure 7d,e). The indicated high gbcxMRS group may have less immunity compared to the low gbcxMRS group. Furthermore, Spearman analysis was performed to comprehensively evaluate the correlation of the biological gene signature reported by previous studies with gbcxMRS. The correlation heatmap demonstrated that gbcxMRS was highly correlated with metabolism and immune signatures (Figure 7f,g; Appendix A).

### 3.7. GbcxMRS Predicted Drug Sensitivity in PDAC

Because chemotherapy is a common method for treating PDAC, we estimated whether gbcxMRS could be used to predict drug sensitives to 5-fluorouracil, irinotecan and docetaxel. We found that the low gbcxMRS group exhibited high sensitivity to 5-fluorouracil treatment. In contrast, the high gbcxMRS group was sensitive to irinotecan and docetaxel treatment (Figure 8a,b). However, the sensitivity of gemcitabine and paclitaxel did not show the significance between gbxcMRS groups (Appendix A). In addition, the subclass mapping results reveal that the high gbcxMRS group may respond to CTLA-4 treatment (Figure 8c,d). Furthermore, the correlations between the 16 genes and the sensitivities to chemotherapeutics were revealed with the 60 cell lines from the NCI-60 database, which illustrated the mechanism of drug sensitivity at the pan-cancer level (Figure 8e). Furthermore, by applying the gbcxMRS into the NCI-60 data, we found that the low gbcxMRS group showed significant drug sensitivity of 5-fluorouracil (Appendix A). These results indicate that gbcxMRS was also valuable in guiding the treatment of PDAC.

## 4. Discussion

The contrast between the incidence and mortality of PDAC exactly illustrates the fatal nature of the disease. Even though an increasing number of advanced diagnostic and prognostic methods are becoming available, the incidence of PDAC is still estimated to be increasing, and 355,317 new cases are predicted in 2040 [3]. Targeting metabolism can be a promising strategy to optimize the prognosis and treatment of PDAC given that the pancreas itself is an organ of physiological metabolism and oxidation [15]. Hypoxia is one of the most important factors for PDAC development and involving the tumor microenvironment [61,62]. Several publications analyzed the PDAC patient data by using the hypoxia pathway independently [63,64]. In addition, we selected the metabolism pathway based on the KEGG metabolism database, in which the hypoxia pathway was not included [65]. In the previous study published by S.R. Rosario et al., the metabolic regulation of pan-cancer based on the TCGA database was comprehensively analyzed. Similarly, they did not include the hypoxia pathway as a metabolism pathway [66]. Therefore, in the present study, we selected and first demonstrated that glycolysis, bile acid metabolism, cholesterol homeostasis and xenobiotic metabolism were crucial metabolic pathways of PDAC and can serve as factors for molecular subtyping. Next, a 16-gene signature for calculating the gbcxMRS was developed via Cox regression and validated in multiple independent external cohorts. We subsequently revealed that gbcxMRS accurately predicted the poor prognostic subtypes of PDAC and was closely related to the ECM and T cells. Meanwhile, the gbcxMRS showed potential efficacy in predicting the recurrence in ICGC and was validated in our own cohort. Furthermore, the gbcxMRS can guide clinical medication decisions.

Our research is of particular significance because the genes used for signature development were derived from crucial metabolic pathways that were cautiously defined based on large-scale cohorts, and the samples included were indeed PDAC, which may be confounded in other studies [67]. Second, five independent external validation cohorts were applied in the present study, which ensured the robustness of the gbcxMRS signature. In addition, we carefully screened the underlying mechanisms and related functions of the signature, which highlighted new prospects of metabolism in PDAC.

Some of the 16 genes in the signature have not been fully studied. Ecdysoneless (ECD) has been studied as a tumor-promoting gene in PDAC that activates pAkt, a molecule that regulates glycolysis in tumor cells and enhances the expression of solute carrier family 2 (facilitated glucose transporter), member 4 (GLUT4) to strengthen glycolysis [68]. The functions of pyruvate carboxylase (PC) have been studied widely in pancreatic beta cells and insulin secretion [69,70]. Recent studies have shown that PC is overactivated in pancreatic cancer cells compared with fibroblasts in the TME, which is necessary for tumor growth [71]. Phosphofructokinase, muscle (PFKM) is a key enzyme in glucose metabolism. Studies have shown that pancreatic tumor cells exhibit low PFKM activity because of cellular O-GlcNAcylation, which leads to KRAS mutations. In addition, studies have shown that *KRAS* mutations promote glycolysis by upregulating the rate-limiting enzymes and transporters of the process, such as lactate dehydrogenase A (LDHA), phosphofructokinase-1 (PFK1) and hexokinase 2 (HK2); thus, there is an interaction between KRAS and glycolysis [18,72]. It was also shown that knockout of PFKM can decrease PD1+ T cells in the TME [73,74]. TPI1, also named ETS proto-oncogene 1 (ETS1), works as a downstream transcription factor of HIF-1a, which increases in a hypoxic environment and is highly correlated with recurrence rate of intrahepatic cholangiocarcinoma patients [75,76]. TPI1 is modulated by miR-381, miR-769-5p and prostaglandin E2, which demonstrates its key role in PDAC [77,78,79]. Cytidine deaminase (CDA) is another independent factor in the present study. Interestingly, CDA can be expressed by both intratumor Gammaproteobacteria and tumor cells, and CDA can degrade gemcitabine, leading to drug resistance [80,81]. Additionally, CDA can induce mutations in KRAS and c-Myc in PDAC [82]. Overall, these studies fully support the reliability of our signature and the genes used for the signature, not only those involved in metabolism but also those associated with the stability of the genome and the TME.

Functional enrichment suggests that the elevation of gbcxMRS was closely related to the organization of the ECM. The ECM has long been regarded as a crucial factor in the progression of PDAC and is composed of laminin, fibronectin, glycoproteins, polysaccharides, etc. [83]. The ECM and stroma can impede the transportation of anticancer agents in vesicles, which leads to chemoresistance. Moreover, the ECM supplies signals for and the degeneration of cell adhesion and metastasis [84,85]. Targeting the ECM has provided new insight into PDAC therapy. Furthermore, a series of studies have indicated the interaction between metabolism and the ECM [86,87]. In this study, we focused more on targeting metabolic pathways to reverse ECM organization in PDAC. High mobility group protein B1 (HMGB1) can upregulate the expression of HIF-1α to induce glycolysis and promote the proliferation of fibroblasts and the formation of ECM [88]. Silencing of glyoxalase 1 (GLO1) can lead to methylglyoxal (MG)-induced dicarbonyl stress to mediate ECM organization [89]. Cholesterol homeostasis is also crucial for the stability of the ECM. Some signaling molecules on the cell membrane should bind to cholesterol or lipid rafts to reduce the activity of tumor cells and impede invadopodia formation by invasive tumor cells in the ECM [90,91]. Studies have also demonstrated that activation of the bile acid receptor Gpbar1 (TGR5) inhibits the degeneration of the ECM, which provides evidence for the importance of bile acid metabolism. However, more profound studies are needed to illustrate the relationship between bile acid metabolism and the ECM [92].

Notably, gbcxMRS has a robust ability to predict the squamous subtype in PDAC (AUC = 89.7%). The squamous type has been reported to be characterized by metabolic reprogramming and the hypoxia response, which was in accordance with the gbcxMRS results in this study [60]. A previous study showed that the squamous type was also significantly associated with recurrence [93]. In addition, our signature accurately predicted the pure-basal-like subtype, another poor prognostic subtype of PDAC (AUC = 86.6%). This pure-basal-like subtype is similar to the basal-like and squamous subtypes, which are resistant to 5-fluorouracil treatment and may potentially respond to immunotherapy [59,94]. The low gbcxMRS showed sensitivity to 5-fluorouracil, which is consistent with this finding. Furthermore, metabolic reprogramming and the exhaustion of T cells have gained increasing attention, and the relationship between metabolism and the immune landscape in PDAC has been partially revealed. Inhibition of hexosamine biosynthesis, a method of bypassing glycolysis, could elevate CD8+ T cell infiltration and enhance anti-PD1 therapy in PDAC [95]. Studies have also indicated that bile acid and glycolysis metabolism has a crucial impact in modulating T helper cells expressing IL-17a (Th17 cells) by derivatives 3-oxoLCA and isoalloLCA [96]. In our data, we observed an increase in Th17 cells in the low gbcxMRS group. It indicated that Th17 cells may be involved in the pro-tumorigenic in the low gbcxMRS group. Similarly, the accumulation of CD8 T cells in low gbcxMRS groups could be one of the reasons why this subgroup of patients has better prognosis. Moreover, it has been reported that regulating cholesterol metabolism enhances CD8+ T cell function and that cholesterol could lead to CD8+ T cell exhaustion. Thus, we demonstrated that cholesterol has a double-edged impact on the TME. It can stabilize tumor cells, but it has an adverse impact on T cells [97,98]. Overall, xenobiotic metabolism reflects the function of the pancreas, which has not been studied in depth to data [99]. Recently, studies have shown that the CTLA-4 blockade does not show the significant response in PDAC [100,101]. However, the metabolism changing, especially the increasing glucose activity, may improve the CTLA-4 therapeutic effect [102]. Moreover, previously papers proved that the CNV (copy number variation) may have a significant effect on immunotherapy, and CNV is also highly correlated with cancer metabolism [103,104]. These could perhaps explain that high gbcxMRS patients may respond to CTLA-4 treatment.

Limitations and further mechanistic exploration should be carried out via basic experiments. Due to a lack of clinical information, the prediction of chemotherapy and immunotherapy should be analyzed by prospective studies in our own cohort in the future.

## 5. Conclusions

In summary, based on the analysis of the overall pathogenesis of PDAC, our study established a robust and reproducible metabolic-related signature, which has outstanding significance for the prognosis and treatment of PDAC.

## Figures and Tables

**Figure 1 cancers-14-01825-f001:**
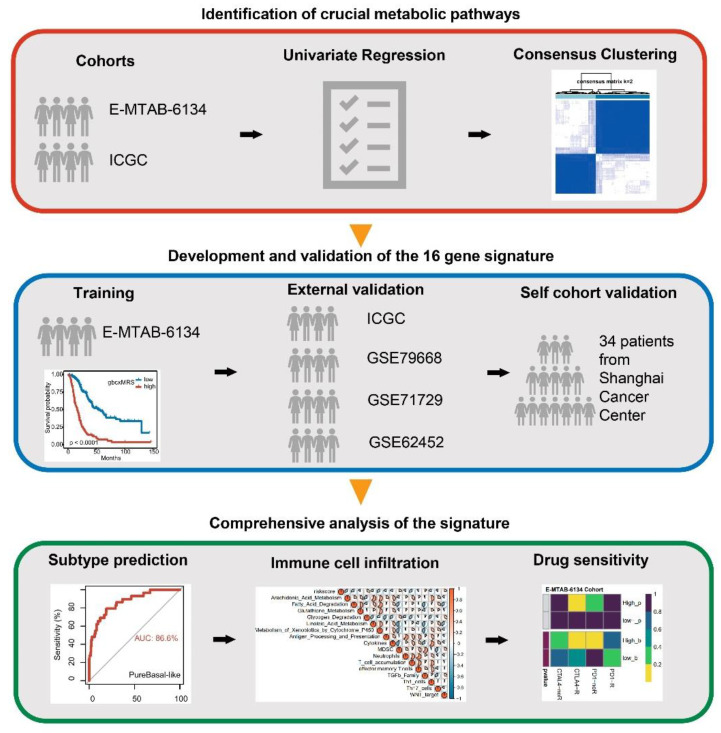
Workflow of the study.

**Figure 2 cancers-14-01825-f002:**
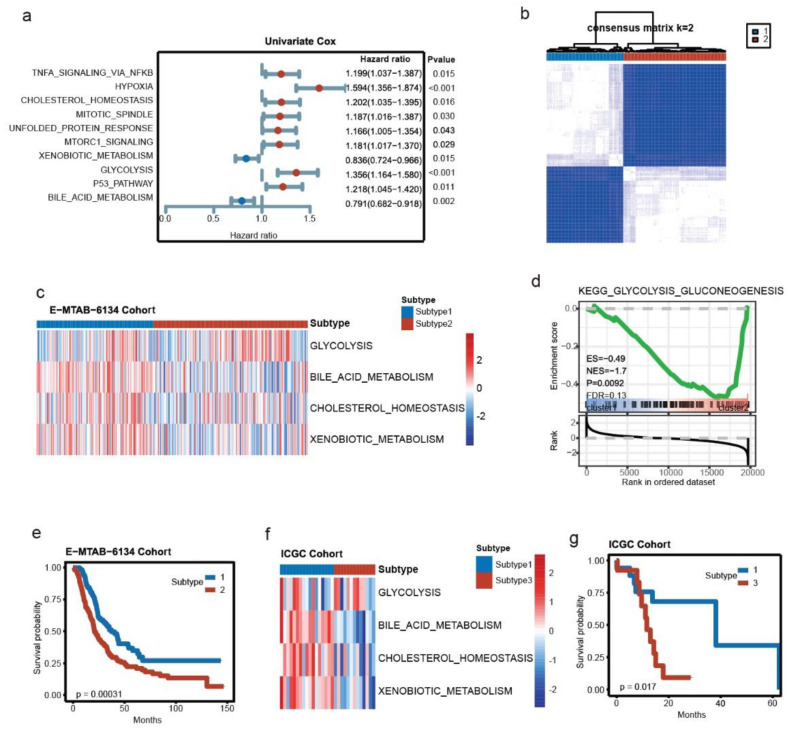
Key metabolic pathways of PDAC and molecular subtyping. (**a**) Univariate Cox regression indicated that glycolysis, bile acid metabolism, cholesterol homeostasis and xenobiotic metabolism were prognostic indicators in PDAC (*n* = 288). (**b**) Two clusters were determined by consensus clustering in the E-MTAB-6134 cohort (*n1* = 123, *n2* = 165). (**c**) Differences in the 4 metabolic pathways between the two clusters in the E-MTAB-6134 cohort. (**d**) GSEA of the glycolysis pathway between the two subtypes in the E-MTAB-6134 cohort. (**e**) Kaplan–Meier survival analysis between the two subtypes in the E-MTAB-6134 cohort. (**f**) Differences in the 4 metabolic pathways between Cluster 1 (*n* = 17) and Cluster 3 (*n* = 13) in the ICGC cohort. (**g**) Kaplan–Meier survival analysis between Cluster 1 and Cluster 3 in the ICGC cohort.

**Figure 3 cancers-14-01825-f003:**
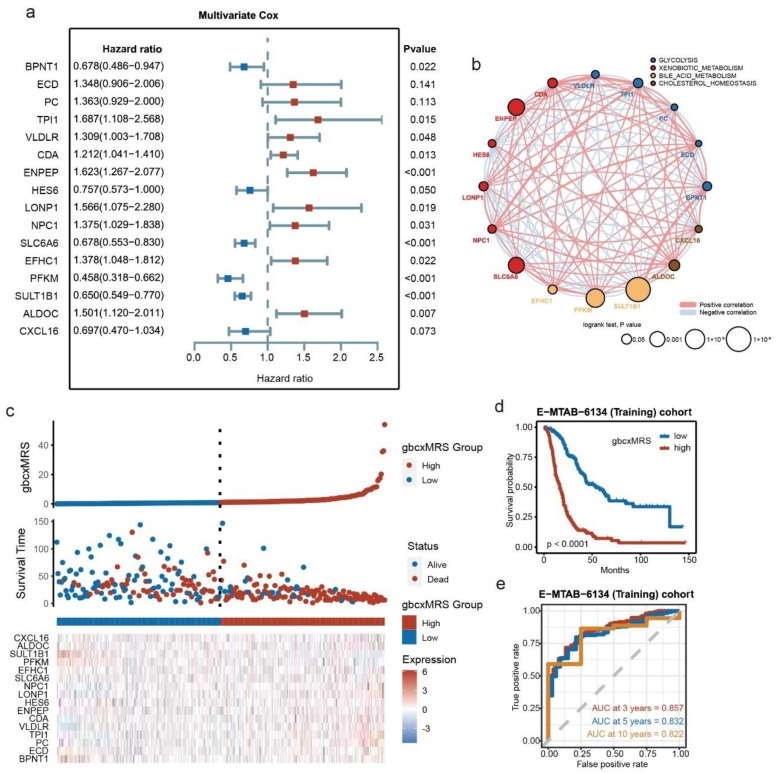
Development of the signature from key metabolic pathways in the E-MTAB-6134 cohort. (**a**) Multivariate Cox regression and stepwise regression finally screened 16 metabolism-related genes and their hazard ratios. (**b**) The network exhibited the pathways to which each gene belongs, the significance of each gene in the model and the correlation between 16 genes. (**c**) The landscape of the gbcxMRS, survival status and gene expression. (**d**) Kaplan–Meier survival analysis between high (*n* = 144) and low (*n* = 144) gbcxMRS groups. (**e**) AUC of 3-, 5- and 10-year time-dependent ROC curves.

**Figure 4 cancers-14-01825-f004:**
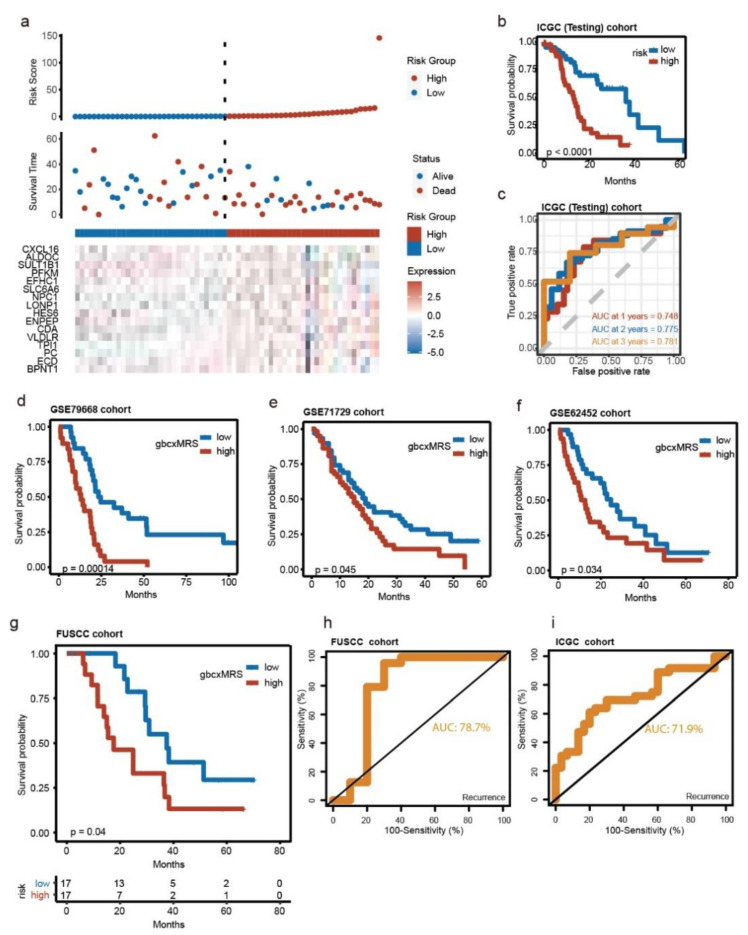
Robust and repeated validation of the 16-gene signature in external cohorts. (**a**) The landscape of the gbcxMRS, survival status and gene expression in the ICGC cohort. (**b**) Kaplan–Meier survival analysis between high (*n* = 33) and low (*n* = 33) gbcxMRS groups in the ICGC cohort. (**c**) AUC of 1-, 2- and 3-year time-dependent ROC curves. (**d**–**f**) Kaplan–Meier survival analysis of high and low gbcxMRS groups in (**d**) GSE79668 (*n* = 51), (**e**) GSE71729 (*n* = 123) and (**f**) GSE62452 (*n* = 65). (**g**) Kaplan–Meier survival analysis between the high and low gbcxMRS groups in the Shanghai Cancer Center cohort (FUSCC). (**h**,**i**) gbcxMRS for predicting the recurrence in FUSCC and ICGC cohorts.

**Figure 5 cancers-14-01825-f005:**
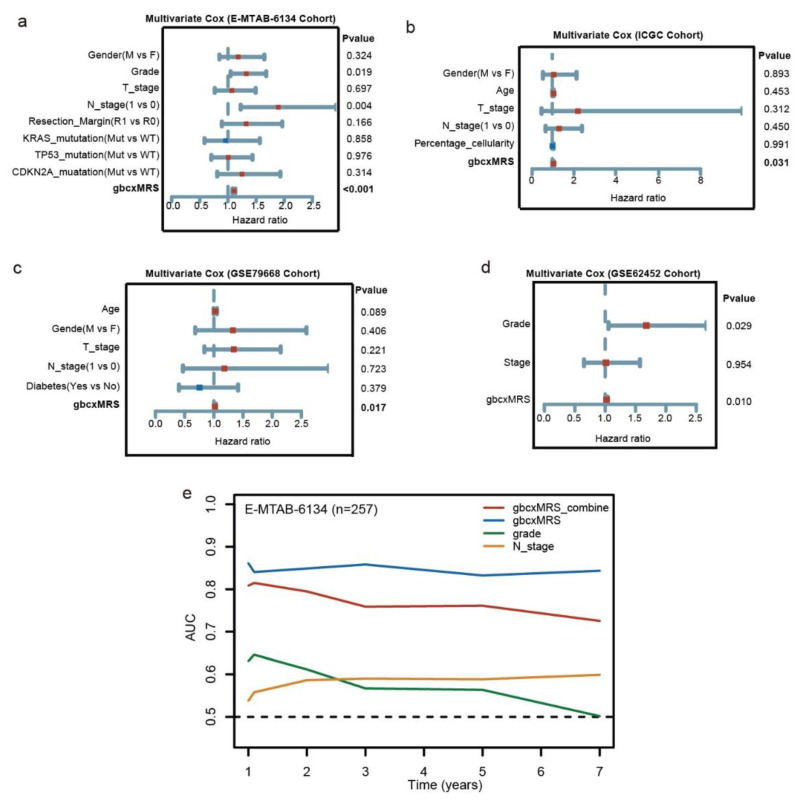
gbcxMRS was an independent and indispensable prognostic factor in PDAC. (**a**–**d**) gbcxMRS was independent of other clinical variants in the (**a**) E-MTAB-6134 cohort (*n* = 257), (**b**) ICGC cohort (*n* = 62), (**c**) GSE79668 (*n* = 51) and (**d**) GSE62452 (*n* = 64). (**e**) AUC of the time-dependent ROC curve of gbcxMRS alone or combined with other clinical indicators in the E-MTAB-6134 cohort.

**Figure 6 cancers-14-01825-f006:**
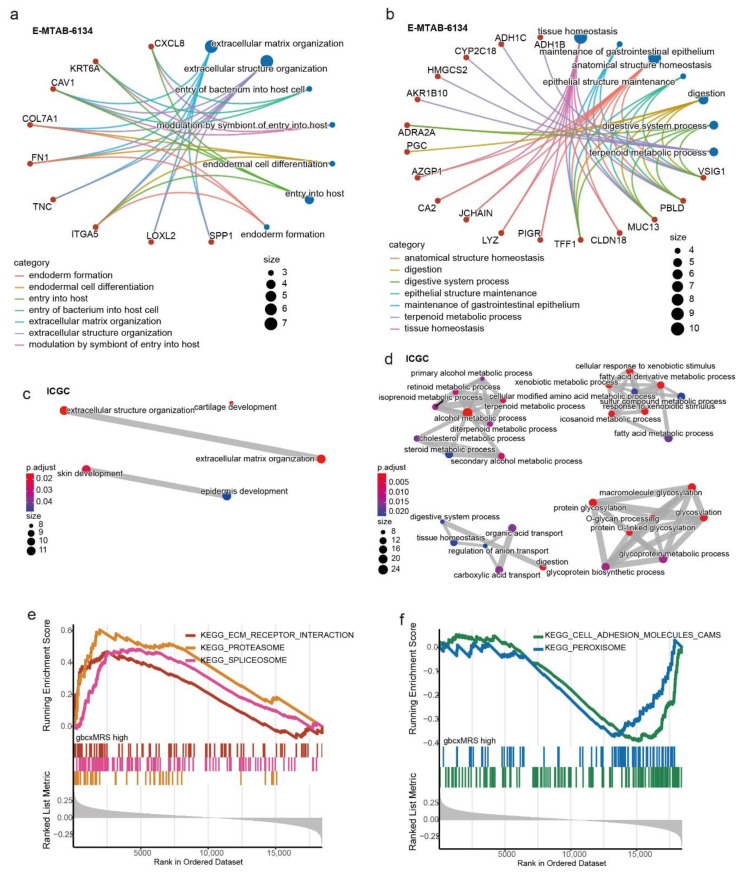
Functional enrichment for the gbcxMRS. (**a**,**b**) GO analysis of the (**a**) upregulated and (**b**) downregulated genes in the high gbcxMRS patients in the E-MTAB-6134 cohort. (**c**,**d**) GO analysis of the (**c**) upregulated and (**d**) downregulated genes in the high gbcxMRS patients in the ICGC cohort. (**e**) Pathways activated in the high gbcxMRS group measured by GSEA in E-MTAB-6134. (**f**) Pathways suppressed in the high gbcxMRS group measured by GSEA in E-MTAB-6134.

**Figure 7 cancers-14-01825-f007:**
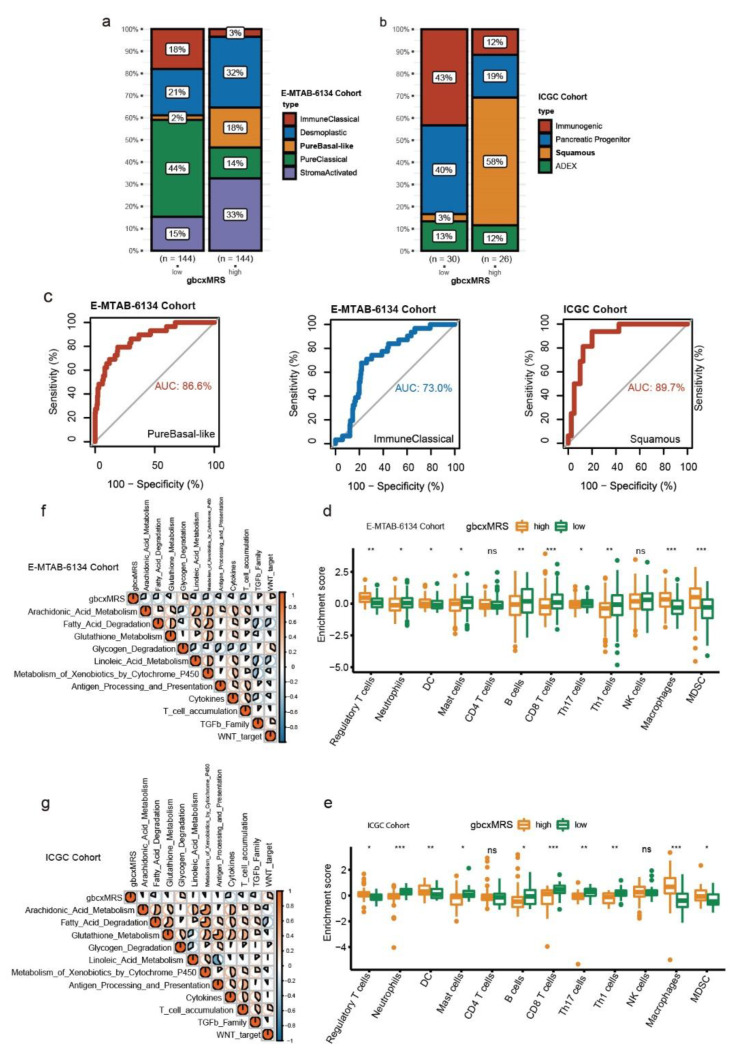
GbcxMRS predicted PDAC subtypes and influenced the tumor microenvironment (TME). (**a**,**b**) The landscape of the proportion of different PDAC subtypes in the high and low gbcxMRS groups in the (**a**) E-MTAB-6134 cohort (*n* = 257) and (**b**) ICGC cohort (*n* = 56). (**c**) gbcxMRS predicted poor prognostic PDAC subtypes with high efficiency. (**d**,**e**) The enrichment score of infiltration immune cells in gbcxMRS subgroups. (**f**,**g**) The relationship between crucial metabolic signatures and immune signatures in the (**f**) E-MTAB-6134 and (**g**) ICGC cohorts. ns, *, **, and *** represent not significant (*p* > 0.05) and significant at the levels *p* ≤ 0.05, *p* ≤ 0.01, and *p* ≤ 0.001.

**Figure 8 cancers-14-01825-f008:**
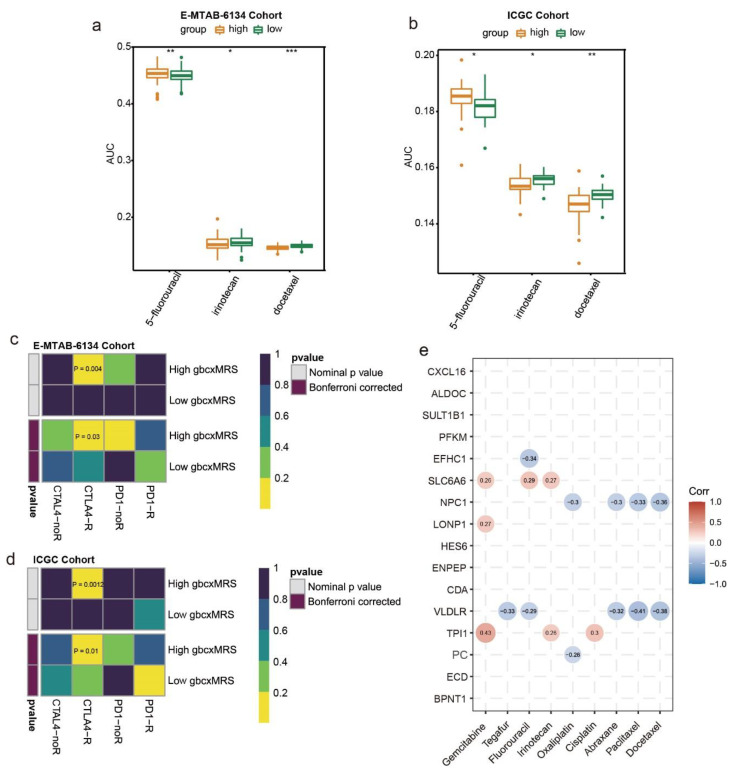
GbcxMRS predicted drug sensitivity in PDAC. (**a**,**b**) The estimated drug sensitivity (log AUC) for 5-fluorouracil, irinotecan, and docetaxel between the high and low gbcxMRS groups in the (**a**) E-MTAB-6134 and (**b**) ICGC cohorts. (**c**,**d**) Submap analysis indicated that the high gbcxMRS group could be more sensitive to CTLA-4 inhibitors in both the (**c**) E-MTAB-6134 and (**d**) ICGC cohorts. (**e**) Sixteen genes were related to the sensitivity of multiple chemotherapy drugs in a pan-cancer analysis using the CellMiner database (27 April 2021). *, **, and *** represent significant at the levels *p* ≤ 0.05, *p* ≤ 0.01, and *p* ≤ 0.001.

**Table 1 cancers-14-01825-t001:** Comparison of clinical features between the high and low gbcxMRS groups in the E-MTAB-6134 cohort.

Clinical Features	gbcxMRS High	gbcxMRS Low	*p* Value
	*n* = 130	*n* = 127	
OS.time	18.5 (14.2)	36.7 (27.7)	<0.001
OS:			<0.001
Alive	23 (17.7%)	72 (56.7%)	
Dead	107 (82.3%)	55 (43.3%)	
Gender:			0.242
Female	48 (36.9%)	57 (44.9%)	
Male	82 (63.1%)	70 (55.1%)	
Grade:			<0.001
G1	38 (29.2%)	65 (51.2%)	
G2	61 (46.9%)	53 (41.7%)	
G3	31 (23.8%)	9 (7.09%)	
T stage:			0.196
T1	3 (2.31%)	8 (6.30%)	
T2	21 (16.2%)	15 (11.8%)	
T3	106 (81.5%)	104 (81.9%)	
N stage:			0.065
N0	25 (19.2%)	38 (29.9%)	
N1	105 (80.8%)	89 (70.1%)	
Resection margin:			0.177
resection margin R0	101 (77.7%)	108 (85.0%)	
resection margin R1	29 (22.3%)	19 (15.0%)	
*KRAS* mutation:			0.093
mutation in *KRAS*	110 (84.6%)	117 (92.1%)	
no mutation in *KRAS*	20 (15.4%)	10 (7.87%)	
*TP53* mutation:			0.14
mutation in *TP53*	96 (73.8%)	82 (64.6%)	
no mutation in *TP53*	34 (26.2%)	45 (35.4%)	
*CDKN2A* mutation:			0.261
mutation in *CDKN2A*	24 (18.5%)	16 (12.6%)	
no mutation in *CDKN2A*	106 (81.5%)	111 (87.4%)	

OS, overall survival; KRAS, Kirsten rat sarcoma viral oncogene homolog; TP53, tumor protein p53; CDKN2A, cyclin-dependent kinase inhibitor 2A.

**Table 2 cancers-14-01825-t002:** Comparison of clinical features between high and low gbcxMRS groups in the FUSCC cohort.

Clinical Features	gbcxMRS Low *n* = 17	gbcxMRS High *n* = 17	*p* Value
Gender			0.084
Female	11 (32.4%)	5 (14.7%)	
Male	6 (17.6%)	12 (35.3%)	
Grade			1.000
High--middle	13 (38.2%)	12 (35.3%)	
Low	4 (11.8%)	5 (14.7%)	
Tissue invasion			0.265
NO	3 (9.4%)	7 (21.9%)	
YES	12 (37.5%)	10 (31.2%)	
Lymph node metastasis			0.084
NO	12 (35.3%)	6 (17.6%)	
YES	5 (14.7%)	11 (32.4%)	
Tumor thrombus			0.688
NO	12 (35.3%)	14 (41.2%)	
YES	5 (14.7%)	3 (8.8%)	
Neural invasion			1.000
NO	2 (5.9%)	1 (2.9%)	
YES	15 (44.1%)	16 (47.1%)	
Recurrence			0.017
NO	8 (23.5%)	1 (2.9%)	
YES	9 (26.5%)	16 (47.1%)	
Age (mean ± SD)	59.82 ± 9.82	61.88 ± 7.51	0.497
Tumor size (mean ± SD)	3.63 ± 1.94	4.06 ± 1.49	0.475
CA19-9, median	51.2 (17.34, 153.2)	448.3 (38.75, 727.1)	0.042
CA125, median	21.06 (15.02, 30.89)	18.69 (11.65, 23.87)	0.357
CA50, median	13.92 (5.41, 83.47)	154.74 (14.65, 327.13)	0.063
CA242, median	7.91 (4.11, 22.47)	52.89 (13.83, 150)	0.025
CEA, median	2.11 (1.71, 2.82)	4.52 (2.67, 7.43)	0.046

CA19-9, carbohydrate antigen 19-9; CA125, carbohydrate antigen 125; CA50, carbohydrate antigen 50; CA242, carbohydrate antigen 242; CEA, carcinoembryonic antigen.

## Data Availability

The datasets generated and/or analyzed during the current study are available from the corresponding author on reasonable request.

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
