# Peer review of "Robust Validation and Comprehensive Analysis of a Novel Signature Derived from Crucial Metabolic Pathways of Pancreatic Ductal Adenocarcinoma"

_cancers, 2022, doi:10.3390/cancers14071825_

Round 1

Reviewer 1 Report

This is a quite comprehensive study.  It is well written and well presented. 

I would request authors to incorporate following  findings:

1. Since Abraxane® (nab-paclitaxel) plus Gemzar® (gemcitabine) is widely used for clinical intervention of PDAC, it will be interesting to find if gbcxMRS can be useful to predict the sensitivity of this combinational therapy.  Authors can extend their analysis for E-MTAB and ICGC cohort to find the sensitivity of Gemcitabine + nab-paclitaxel.

2. Authors presented  drug sensitivity data of 60 cell lines (NCI-171 60)  downloaded from CellMiner. It will be very useful to incorporate gbcxMRS analysis of these cells to confirm the alignment of findings with patient data.

Author Response

Response to Reviewer 1:

We sincerely appreciate your time and efforts in reviewing our paper. We have learned a great deal from your comments, and we have revised our manuscript as suggested.

  1. Since Abraxane® (nab-paclitaxel) plus Gemzar® (gemcitabine) is widely used for clinical intervention of PDAC, it will be interesting to find if gbcxMRS can be useful to predict the sensitivity of this combinational therapy.  Authors can extend their analysis for E-MTAB and ICGC cohort to find the sensitivity of Gemcitabine + nab-paclitaxel.

Answer: Thank you for your kindness suggestions. The gemcitabine and paclitaxel are widely used in the clinical for PDAC treatment.  However, based on the database of CTRP v2.0 and PRISM, there are no such data supporting us to predict the sensitivity of combination treatment of gemcitabine + paclitaxel. Therefore, we performed the drug sensitivity analysis in gemcitabine and paclitaxel alone, but the result showed that there was not significant difference between gbcxMRS group. Please see the Figure S4a b. We added this result into the Results section. For details, please see page 19, line 320-321.

  1. Authors presented drug sensitivity data of 60 cell lines (NCI-171 60) downloaded from CellMiner. It will be very useful to incorporate gbcxMRS analysis of these cells to confirm the alignment of findings with patient data.

Answer: We are grateful for your valuable advice. We applied the gbcxMRS to the cell lines. And the results showed that the IC50 of 5-fluorouracil has significantly lower in the gbcxMRS low group. It is consisted with our data that low gbcxMRS in PDCA patients may have good response with 5-fluorouracil treatment. We revised the Materials&Methods and Results section. Please see page 5, line 175-176; page 19, 326-327 and Figure S4c.

Reviewer 2 Report

This is a very nice study and truly robust in the extent of analysis.

  1. Once the authors have decided to use the acronym PDAC, I would advise them to continue to use it throughout the manuscript. Please avoid using pancreatic cancer, or pancreatic adenocarcinoma (line 133). The message that the current data only relates to PDAC must remain consistent
  2. The problem with transcriptome data is the admixture of NETs. I am so pleased to see the rigour of the analysis.
  3. The authors need to acknowledge and cite previous published work highlighting the significance of metabolic pathways to the management and prognosis of PDAC: PMID: 28163917, PMID: 34514690 and PMID: 30677401
  4. Tables 1 & 2, please provide a list of abbreviations, eg OS, and the individual genes, tumour markers, etc.

Author Response

Response to Reviewer 2:

We respect your suggestions tremendously; they have helped us to improve our work a great deal. We have endeavored to revise our manuscript based on your comments and would be open to further advice.

1.Once the authors have decided to use the acronym PDAC, I would advise them to continue to use it throughout the manuscript. Please avoid using pancreatic cancer, or pancreatic adenocarcinoma (line 133). The message that the current data only relates to PDAC must remain consistent

Answer: We appreciate you for highlighting this error. We changed the pancreatic adenocarcinoma and some “pancreatic cancer” to PDAC following your suggestions. Please see page,2, line 98; page 3, line 104-105 and line 132; page 5, line 172; page 20, line 332 and 334; page 22, line 439.

2.The problem with transcriptome data is the admixture of NETs. I am so pleased to see the rigour of the analysis.

Answer: We are grateful for your valuable advice. Accordingly, we the only enrolled the PDAC patients for the analysis. The criteria are based on the clinical information that patients whose pathological type was PDAC were included. And we mentioned in Materials and Methods section (2.1 Patient cohort). Therefore, NETs have been excluded for further analysis. For further details, please see page 3, line 123-124 and line127-128 (with highlight).

3.The authors need to acknowledge and cite previous published work highlighting the significance of metabolic pathways to the management and prognosis of PDAC: PMID: 28163917, PMID: 34514690 and PMID: 30677401

Answer: Thank you for your suggestion. We added the reference according to your suggestions. Please see page 22, line 495-497 and line 511-516.

4.Tables 1 & 2, please provide a list of abbreviations, eg OS, and the individual genes, tumour markers, etc.

Answer: Thank you for your comments. We added the abbreviations for table 1 and table 2. Please see page 12, line 264-265 and page 14 line 269-270.

Reviewer 3 Report

The authors performed the systematic analysis to explore gene set which involving in regulation of glycolysis, bile acid metabolism, cholesterol homeostasis and xenobiotic metabolism. Their finding may accelerate the repurposing of 'old' drugs to treat PDAC. Despite this manuscript is informative, the functional analysis is not based on experiments. Moreover, I find that the authors seem to skip the hypoxia from their univariate Cox regression analysis. The authors  should provide the rational explanation for skiping hypoxia.

Author Response

Response to Reviewer 3:

We thank you for your effort in reviewing our manuscript. Please see our response to your comments below.

The authors performed the systematic analysis to explore gene set which involving in regulation of glycolysis, bile acid metabolism, cholesterol homeostasis and xenobiotic metabolism. Their finding may accelerate the repurposing of 'old' drugs to treat PDAC. Despite this manuscript is informative, the functional analysis is not based on experiments. Moreover, I find that the authors seem to skip the hypoxia from their univariate Cox regression analysis. The authors should provide the rational explanation for skiping hypoxia.

Answer: Thank you for your kind suggestions and comments. Hypoxia is one of the most important factors for PDAC development and involving with tumor microenvironment. Several publications analyzed the PDAC patients’ data by using the hypoxia pathway independently. In addition, we selected the metabolism pathway based on The Kyoto Encyclopedia of Genes and Genomes (KEGG) metabolism database which hypoxia pathway is not included. The previous study published by S.R. Rosario et al., they comprehensively analyzed the metabolic regulation in Pan-cancer based on TCGA database. Similarly, they didn’t include the hypoxia pathway as a metabolism pathway. Furthermore, we have supplemented a short discussion in Discussion section. Please see page 20, line 338-345.

Round 2

Reviewer 1 Report

Authors have revised the manuscript as requested.

Reviewer 3 Report

The authors have provided appropriate explaintations and let this manuscript become more better. This revised manuscript seems to reach the criterion for publish.